# Identifying Cross-Utilization of RSV Vaccine Inventions across the Human and Veterinary Field

**DOI:** 10.3390/pathogens12010046

**Published:** 2022-12-27

**Authors:** Marga Janse, Swasti D. Soekhradj, Rineke de Jong, Linda H. M. van de Burgwal

**Affiliations:** 1Athena Institute, Faculteit der Bètawetenschappen W&N Gebouw, VU Amsterdam, De Boelelaan 1085, 1081 HV Amsterdam, The Netherlands; 2Wageningen Bioveterinary Research, Houtribweg 39, 8221 RA Lelystad, The Netherlands

**Keywords:** vaccine development, vaccine inventions, collaboration, knowledge-sharing, HRSV, BRSV

## Abstract

The respiratory syncytial virus (RSV) has two main variants with similar impact, a human and a bovine variant. The human respiratory syncytial virus (HRSV) is the most frequent cause of acute respiratory disease (pneumonia) in children, leading to hospitalization and causing premature death. In Europe, lower respiratory tract infections caused by HRSV are responsible for 42–45 percent of hospital admissions in children under two. Likewise, the bovine respiratory syncytial virus (BRSV) is a significant cause of acute viral broncho-pneumonia in calves. To date no licensed HRSV vaccine has been developed, despite the high burden of the disease. In contrast, BRSV vaccines have been on the market since the 1970s, but there is still an articulated unmet need for improved BRSV vaccines with greater efficacy. HRSV/BRSV vaccine development was chosen as a case to assess whether collaboration and knowledge-sharing between human and veterinary fields is taking place, benefiting the development of new vaccines in both fields. The genetic relatedness, comparable pathogeneses, and similar severity of the diseases suggests much can be gained by sharing knowledge and experiences between the human and veterinary fields. We analyzed patent data, as most of pharmaceutical inventions, such as the development of vaccines, are protected by patents. Our results show only little cross-utilization of inventions and no collaborations, as in shared IP as an exchange of knowledge. This suggests that, despite the similarities in the genetics and antigenicity of HRSV and BRSV, each fields follows its own process in developing new vaccines.

## 1. Introduction

The respiratory syncytial virus (RSV) has two main variants with similar impact, a human and a bovine variant. The human respiratory syncytial virus (HRSV) is the most frequent cause of acute respiratory disease (pneumonia) in children, leading to hospitalization and causing premature death with a 15% mortality rate in children under the age of five. Generating over 34 million cases each year, HRSV infections cause more than 90% of children’s mortality in low- and middle-income countries [1,2,3,4,5]. In Europe, lower respiratory tract infections caused by HRSV are responsible for 42–45% of hospital admissions in children under two [6]. While the direct clinical impact of RSV infections on the global community is comparable to that of influenza, the indirect socio-economic burden is far greater. Negative health outcomes of HRSV include respiratory complications, short- to long-term declining lung function, a 2.7 times higher risk of developing pneumonia, wheezing, and asthma. In the US, asthma and wheezing cause long-term financial and social burdens accounting for 10–18% of the use of health care resources [7]. Likewise, bovine respiratory syncytial virus (BRSV) is a significant cause of acute viral broncho-pneumonia in calves [8]. BRSV infections in the cattle industry are characterized by significant morbidity and mortality, leading to economic losses and costs for treatment and preventions. Subsequently, these costs lead to a loss of production and the reduced value of the animal [9,10]. Additionally, the severity of respiratory syncytial virus (RSV) infections differs with age in both humans and bovines, with primary infections running a more severe course when contracted at a younger age.

Vaccination is an important means of preventing and controlling infectious diseases. Large-scale immunization efforts through vaccination can be considered one of the greatest contributors to global health of any human intervention [11]. Edward Jenner’s experiments with a smallpox vaccine in 1796 ultimately led to the eradication of the disease in 1979 [12]. Other success stories include an over 95% global incidence reduction of measles, mumps, rubella, pertussis, tetanus, and diphtheria [13]. More recently, the COVID-19 vaccinations in the US have resulted in an estimated reduction of 140,000 deaths in the first 5 months of the vaccination campaign in 2021 [14]. According to the estimates of the World Health Organization (WHO), immunization efforts currently prevent 2–3 million deaths per year, and this is achieved in a remarkably cost-effective way [15]. Vaccines have not been successfully developed for all types of infectious diseases, however, and to date a large unmet need remains for a safe and effective HRSV vaccine that can be used in young children. These children have not yet developed a fully matured immune system, which makes it challenging to develop an effective vaccine. Despite a high burden of the disease, worldwide spread of the disease, and extensive research over the years, no licensed HRSV vaccine has been developed yet [16,17]. Moreover, after severe safety issues with formalin-inactivated HRSV vaccine candidates developed in the 1960s [18], the innovation progress has been hampered substantially. Remarkably, this is in stark contrast to BRSV vaccines, which have been on the market since the 1970s, as minor risks are acceptable in livestock. Even so, there is still an articulated unmet need for improved BRSV vaccines, specifically ones with a greater efficacy in the presence of BRSV-specific maternal antibodies and in the immunologically immature host. Additionally, there is an unmet need for vaccines with increased potency to induce long-term protection beyond the 6–12 months induced with current vaccines [19,20,21].

According to the One-Health (OH) perspective, the health of animals, humans, and the environment is intertwined, and therefore various infectious diseases occurring across human–animal species barriers need to be addressed together. Human and bovine RSV, although genetically closely related, are not known to be zoonotic, but both are highly infectious and prevalent around the globe [22,23]. The genetic relatedness, comparable pathogeneses, and similar severity of the disease in elderly, young children, and young calves suggest much can be gained by sharing knowledge and experiences between the human and veterinary fields. On a more strategic level, collaboration between the human, animal, and environmental fields is slowly increasing, and networks of professionals in the fields of health and environment have been created (e.g., Med-Vet-Net, ArboZoonet, and NBIC) in order to bridge the communication gap between science and society [24]. The need for intensified collaboration between these fields is rapidly increasing due to the rise of human infectious diseases with a zoonotic origin, like COVID-19, and the increase of antibiotic resistance threatening (global) public health [25]. Whether this intense strategic collaboration is also translated into more sharing of knowledge and inventions on the specific pathogen-level, however, is unclear. As both the human and veterinary field are developing RSV vaccines, it is of interest to assess whether this case, which *prima facie* harbors close alignment in unmet needs, demonstrates intensified One-Health collaboration.

In this paper, we therefore aim to gain insight into whether and to what extent the cross-utilization of RSV innovations and collaboration across the human and veterinary fields for the purpose of RSV vaccine development is occurring. To this purpose, a dataset containing patent documents concerning inventions for the development of RSV vaccines was constructed and analyzed.

## 2. Materials and Methods

HRSV/BRSV vaccine development was chosen as a case to assess whether collaboration and knowledge-sharing between human and veterinary fields is taking place, benefitting the development of new vaccines in both fields. For this purpose, we analyzed patent data, as the majority of all pharmaceutical inventions, like the development of vaccines, are protected by patents, offering solid intellectual property rights for novel pharmaceutical substances. Patent data is one of the most used sources of data for interpreting and evaluating technical progress, since they are: generalizable, empirical, qualitative, and quantitative [26,27]. The qualitative and quantitative analysis of patent documents is used to bridge the gap between research activities as documented in scientific journals and practical applications as found in practice, hence making patent applications indicators of early-stage innovation output [28]. The aim of the patent analysis was to assess the trends and breakthroughs in RSV vaccine development over the period 1995–2020 to identify the extent of the cross-utilization of innovative inventions and collaborative efforts across the human and veterinary fields.

### 2.1. Data Collection and Selection

Patent documents were retrieved from the European Patent Office (EPO) Espacenet database with the guidance of an expert from the Dutch Patent Office (Dutch RVO). The cooperative patent classification (CPC) system was used to generate search criteria to identify all advancements in the human and veterinary fields connected to RSV vaccine development. After selecting relevant CPC codes, the codes were combined with keywords acquired from the literature. Step by step, the search query was expanded, and the resulting patents were assessed on relevance based on the title and abstract of the patents that were applied for between January 1995 and January 2020. Table 1 shows the keywords that were matched to CPC-codes with the use of Boolean operators.

### 2.2. Data Analysis

From Espacenet, 2375 patent documents were retrieved and exported into one comprehensive dataset in Microsoft Excel. All patent documents were deduplicated based on patent family (i.e., multiple jurisdictional filings related to a single invention), reducing the dataset to 1486 patent documents. By performing a manual analysis of all patent documents on title and abstract and applying the inclusion and exclusion criteria as listed in Table 1, 314 patent documents were deemed significantly related to RSV vaccine development for patent analysis. The 314 patent documents were imported into Atlas TI for a qualitative full-text analysis, leading to the elimination of two more documents. In this qualitative full-text analysis, patents were coded as mentioning specific target groups (i.e., human, veterinary, or both) and as being built upon specific technology types, including nucleic acid, whole-virus-inactivated, and subunit antigens.

To understand the potential impact of the inventions, the target markets of the patented inventions were analyzed. First, the application trends of patents were analyzed and depicted in Figure 1. Given the high costs related to patenting, there is a direct relation between the location of patent applications and target markets: patents are generally only applied for in countries where they are expected to bring a substantial economic benefit, either by generating revenues or by preventing competitors from entering the market [26]. Patent families for which only documents with kind codes belonging to the World Intellectual Property Organization (WIPO) or European Patent Office (EPO) were categorized as a separate group. Years of application were indicated, and the countries of patent applications were determined by the country codes of each patent, including all patent applications listed as “also published as”, indicating the publication of documents in specific jurisdictions for which patent protection is also applied. The geographical information (in % per target group) for the different target fields was analyzed and is depicted in Figure 2. Analysis of the stakeholder type enhances the interpretation of the relevance of the data for the target market. To this purpose, unique patent applicants were identified, and patent applicants were categorized as academia, industry, or government key applicants. Applications of patent documents by individual researchers or unknown applicants were categorized as other (Figure 3). It soon became clear that industry holds the most patents; Figure 4 shows the distribution in the industry companies. Lastly, nine categories were created describing the innovative inventions (Figure 5).

We searched for the co-ownership of patents between the different applicant categories (industry, academia, government, and others). Considering co-ownership or co-patenting as the phenomenon of more than one organization contributing to an invention, for which at least the sharing of information on R&D collaboration and IP sharing arrangements is conclusive.

As an indicator of knowledge-sharing and collaboration, co-ownership is conducive to a higher quality of patents and thus strengthens the position of the applicants [29]. Technology and inventions are not static, and new inventions are filed every day. When inventions from earlier patents are cited in later patents, we consider that a “forward citation” [30]. The frequency of the citations of the original patent in new patents shows how important the invention is. Assessing the cross-knowledge-sharing and the impact of the use of new technologies, the number of successor patents was used as a metric to evaluate the knowledge-sharing and the quality and value of the inventions described in the original patent documents [29,30,31].

## 3. Results

### 3.1. Application Trends

For the final dataset of 312 patent documents, patent application trends were visualized for the period 1995–2020 for three target fields: for human use, veterinary use, and the combination of human and veterinary use. Human–veterinary use was defined when patents mentioned both human and veterinary target groups. The timelines (Figure 1) illustrate an overall increase in the number of patent applications for human use. In contrast, the patent applications for human–veterinary use and veterinary use are only slightly increasing and are stabilizing.

### 3.2. Geographical Distribution of the Applications

Asia shows the largest % of patent applications for all three target groups, followed by EA/WIPO/EPO and North America (Figure 2). Europe shows the highest proportion of patent applications for veterinary use and Asia and North America for human use. These results suggest that the Asian and North American markets are considered the most interesting markets for HRSV vaccines, playing an important role in the research and development of these vaccines. This is in contrast to Europe, which is deemed the most interesting market for the development of BRSV-vaccines. Using EA/WIPO/EPO, as application points suggests that the applicants want to secure their invention in more than one country/region.

### 3.3. Applicants

The patent applicants were categorized as industry, academia, government, and other. Overall, industry applicants were most prolific in all three target fields accounting for 58% of the total number of patent documents (see Figure 3), followed by academic applicants.

Two-thirds of all patents intended for human use are filed by large pharmaceutical companies, such as Janssen Vaccines & Prevention owning the largest number of 19 patents, followed by Medimmune, GlaxoSmithKline, and Aventis Pasteur all owning 10 patents each (Figure 4). In the human–veterinary target field, PF Medicament owns 14% of patents. Merial Sas (now Boeringher Ingelheim) holds 13% of the patents in the veterinary field. Notably, only Novavax applied for patents in all three target fields (See Figure 4).

### 3.4. Technologies

In Figure 5, the technological distribution of RSV inventions is depicted for the three target fields. The patent documents contained several different technologies: live attenuated, protein-based (including whole-virus inactivated, particle-based, and subunit sub-categories), nucleic acid (DNA, mRNA), gene-based vectors, combination vaccines, adjuvants, and the production or purification of RSV vaccines. For all three target fields, the technology for subunit vaccines using RSV F proteins is the most described innovative invention in the patent documents.

Overall, the patents for human use contain more variation in technological inventions than the patents for veterinary use. The technological distribution of vaccine platforms found in the patents for human use are subunit antigens (41%), nucleic acid (DNA and mRNA based, 16%), particle-based (12%), gene-based vectors (10%), combination vaccines (9%), live attenuated (8%), adjuvants (2%), whole-inactivated (1%), production RSV (1%). Gene-based vectors such as adenovirus were found only in human patent applications.

The patents for veterinary use, in comparison to patents for human and human–veterinary use, contained the following technological findings: subunit antigens (26%), live attenuated (26%), combination vaccines (22%), whole virus inactivated (13%), and nucleic acid (DNA-based, 13%). The patents for human–veterinary use contained technological findings combining the human and veterinary fields, leaning more to the human field in terms of the use of vaccine platforms. The distribution of vaccine platforms found was subunit antigens (38%), nucleic acid (DNA based, 31%), particle-based (17 %), live attenuated (7%), combination vaccines (4%), and adjuvants (3%).

### 3.5. Strains

The qualitative study of the patent documents identified the use of homologous RSV strains for the corresponding target fields: human, veterinary, and human–veterinary. Most of the human patent applications focused on human HRSV strains of antigenic subtype A rather than strains of subtype B. The veterinary patent applications describe the use of BRSV strains of subtype A (375, 391, 51908) and strain 2335 (subtype not found). Most strains found in the human–veterinary patents were applicable to the above-mentioned human and bovine strains.

### 3.6. Co-Ownership

By analyzing the co-patenting applications for human use, 34 occurrences of co-ownership by 32 applicants of patents were found. Analyzing the type of applicant shows that 49% of the co-ownership is between applicants from American industry. Followed by 24% co-ownership occurring between European applicants from industry and academia, 9% between Chinese applicants from academia and industry, and 3% between Australian applicants from academia and industry. In the human patent field, only 15% of the co-ownerships are over cross-country boundaries, showing the most interactions between American applicants and Canadian and Chinese applicants. For European applications, no co-ownerships were seen with other applicants based in the EU, suggesting country boundaries are a factor influencing collaboration between applicants. Of all co-ownership in the human field, 18% took place between industry and academia suggesting that cooperation in knowledge-sharing leading to innovation between industry and academia is present but limited. For the veterinary patent applications, only one co-ownership was found for two American companies: Wyeth Corporations and American Home Products. In the human–veterinary patent field, co-ownership was only found in academia settings. The Ohio State Innovation Foundation, Research Institute at Nationwide Children’s Hospital, U.S. government, and University of Florida are applicants that collaborate in the human and human–veterinary patent fields. No co-ownership of patent applications was found for applicants usually applying for patents for veterinary use or human use, suggesting there is no knowledge-sharing between applicants focused on different target fields.

### 3.7. Follow-Up of Innovative Inventions

In 129 new successor patent documents, (forward) citations were found with origins in the human and human–veterinary patent fields, suggesting the mutual use, knowledge-sharing, and valorization of the patent content concerning many different technologies. Thereafter, in 12 new successor patent documents, citations were found originating from the human and veterinary patent fields also concerning a broad variety of technologies. In only three new successor patent documents, citations were seen deriving from the veterinary and human–veterinary fields, which were the lowest in citations concerning animal models. Finally, in only five new successor patent documents citations were seen deriving from patents in all three fields. Technology developments are not static, and new patent applications are filed every day. The field of citation documents grows proportionally, and later applications can always refer to previously submitted applications. This makes this patent overview a snapshot for the chosen time frame and not a fixed overview [30].

## 4. Discussion

Herein, we show an increasing number of patent applications for human RSV-vaccine development. In contrast, the applications for veterinary RSV vaccine development and patents with inventions targeting both fields (human–veterinary) showed only a small increase, stabilizing over the last two decades. Although applicants from both the human and the veterinary fields are working on inventions for the development of vaccines in both fields, no co-ownership of patents between the applicants for veterinary use and human or human–veterinary use was found. Next to this, in only a few subsequent patent documents, citations were found originating from both the human and veterinary patent fields. Overall, this suggests that only little participation in R&D collaboration and IP knowledge-sharing is occurring between the different target fields.

Looking at the extensive patenting for human use in Asia and North America suggests that these regions are considered to offer the best market opportunities for bringing a human RSV vaccine to the market. To date, the licensing of a safe and effective HRSV vaccine has not yet been achieved, but the increasing patent activities suggest that RSV-research is extensive [17,32]. In recent years, over 60 RSV vaccine candidates were at various stages of development, ranging from early nonclinical studies to Phase 3 trials in adults and including maternal RSV vaccination as an alternative strategy to provide neonatal protection for young infants [33,34]. Recently, first steps were taken to investigate how to move forward from clinical trials in adults to RSV-naïve infants, based on the expectation that multiple products aimed for pediatric use will soon be ready to enter this final stage of evaluation [16,35].

The stabilizing trend of patent applications for veterinary use and for human–veterinary use suggests that the field has reached a stage of technological limit or saturation [36]. However, to meet the continuing unmet need for HRSV vaccines and to improve currently commercially available BRSV vaccines, R&D activities are essential [20,21,37]. The greatest challenges, for both human and animals, is to elicit a strong, balanced, and stable immune response at a very early age when infants and calves are at the greatest risk of being severely affected by RSV infection. Looking at current R&D activities and the resulting patent applications, three technologies stand out that are frequently applied for in all three fields: live attenuated, sub-unit antigen-based and nucleic-acid-based vaccines. The results clearly indicate that the veterinary field continues to rely on the technology of live attenuated vaccines, as a reliable and cost-effective method of vaccination. This is in line with the commonly shared idea that veterinary disease control must be cost-efficient [38]. Initially, immunization in humans with virus-inactivated vaccines resulted in enhanced disease progression after natural infection in vaccinated infants. This led to serious concerns about the safety and discontinuation of this vaccine technology for use in the human field. Recently, new attenuated strains have been generated for human use, but it proved difficult to select live candidates with the right combination of safety and immunogenicity. Only one new candidate strain revealed to be safe, moderately immunogenic, and genetically stable for infants and children, aged 6–24 months old [39].

The prominent patent applications for the development of human subunit-vaccines are due to new insights into structural aspects of the viral proteins, allowing the design of a vaccine with a conformation close to their native viral form, leading to the development of several subunit vaccines [40]. For the veterinary field, recently two sub-unit vaccine candidates were found to be safe and effective for one-month-old calves with maternally derived serum antibodies [41], suggesting that the subunit-vaccine technology is promising for the development of vaccines in both fields. The development of vaccines using nucleic acid technology is found in all fields, with mRNA inventions only being used in the human field. The colossal investments; the extraordinary efforts of the scientific and medical communities testing the many vaccine candidates; the willingness of BioNTech, Pfizer and Moderna to take risks; the flexibility of regulatory agencies to allow the simultaneous conduction of clinical trials all contributed to speeding up the process of bringing mRNA-based SARS-CoV-2 vaccines to the market [42,43]. This suggest that the development of mRNA-based vaccines for the veterinary field is still far too costly. This also seems to be the case for gene-based vector vaccines, an upcoming technique in the human field inspired by using recombinant adenovirus vector-based vaccines as an example of being well suited to address existing technology [44]. For human (and human–veterinary) use, particle-based vaccine technology is one of the novel technologies that is increasingly seen as the next generation of nanoparticles in the field of vaccination. The advances of nanoparticle design and better knowledge of viruses (structure and mechanism of cell entry) offer strong opportunities in the near future to develop nanoparticles with additional features for RSVs [45]. Recently, promising results with regard to safety and immunogenicity for maternal vaccination with HRSV F-nanoparticles vaccines have already been reported [34]. No patent applications for particle-based and vector vaccines were found for use in the veterinary field, however.

Nevertheless, the genetic and antigenic close relatedness between the human and bovine virus creates opportunities to translate the technology described in patent documents for human RSV to bovine RSV and apply this in the bovine species, as has been described in recent experimental settings [41,46]. Relevantly, young calves offer a valuable pre-clinical model to evaluate the clinical efficacy of vaccines containing human RSV protein(s) that induce antibodies cross-reacting with bovine RSV. In this model, bovine RSV is used to infect HRSV-vaccinated calves to reproduce a natural course of the infection characterized by respiratory disease symptoms that are highly similar to the lower respiratory tract symptoms observed in primary infected infants [39,47].

The patent documents for human–veterinary use mention inventions suitable for both the human and veterinary fields, suggesting a connection between the fields. However, when looking more in depth into the co-ownership of the patents and cross-over citations between the patents for veterinary and human/human–veterinary use, this is hardly the case. The absence of co-ownership of patent documents between these fields suggests that the inventions described in patents for human or human–veterinary use is not of value to the veterinary field and vice-versa. Only a few co-owned patents for human use were found applied for by US industry and US academia. This indicates a limited contribution from the early developments arising from academic R&D and little progress in the market of the inventions by industry [48]. However, the applicants for most of the patents for human use are pharmaceutical companies, especially those known to produce successful vaccines for human use, being Janssen Vaccines & Prevention, owner of the largest number of human patents; Medimmune; and GlaxoSmithKline, suggesting that industry has confidence in future developments and are ready to go to market whenever there is an effective and safe vaccine for human use. Next to this, the significant cross-over use of information in subsequent patents was found to stem from previous patents describing inventions for human/human–veterinary use, suggesting that human vaccine development is expanding. This in contrast to the little cross-over use of information from patents for both veterinary and human use found in a small number of successor patents, mostly describing techniques fit for both fields, suggesting that technological knowledge is being exchanged but only on a small scale.

## 5. Conclusions

The aim of this research was to assess whether and how knowledge is exchanged between the veterinary and human research fields in case of their shared unmet need for developing reliable and safe RSV vaccines in both fields. In conclusion, only little cross-utilization of inventions and no collaborations, as in shared IP as exchange of knowledge, were found between the human and veterinary field. This suggests that, despite the similarities in the genetics and antigenicity of HRSV and BRSV, each field follows its own process in developing new vaccines. The OH perspective could be a helpful approach for the development of vaccines in both fields, but this seems to be an underutilized approach. A common OH goal supports the de-categorizing of the human and veterinary fields in RSV vaccine development, which is needed to facilitate collaboration between the fields [49]. Adopting the OH approach has the potential to save time and money by strategically leveraging the virus similarities and sharing newly created knowledge between both fields. The development of sub-unit vaccines seems to be a possible inventive technique for sharing knowledge and experiences between both fields [17]. Promoting collaboration between a wide diversity of stakeholders in general, however, is a challenging aspect in the vaccine development process and even more challenging when it comes to working from an OH approach, encountering multiple challenges, such as the funding, policy, surveillance, defining, and execution of the OH approach [50]. Compounding this complexity are the differences in target groups (e.g., young calves and infants but also elderly people with health problems), willingness to pay (which is much higher in the human field than the veterinary field), and sensitivity to safety concerns. To understand whether the potential knowledge-sharing and collaboration between the human and veterinary fields, as a contribution to solving the unmet need for safe and effective vaccines for both fields, is encountering challenges hampering collaboration, we recommend a follow-up study asking experts, from both fields, for their perception on the challenges they encounter involving collaboration.

## Figures and Tables

**Figure 1 pathogens-12-00046-f001:**
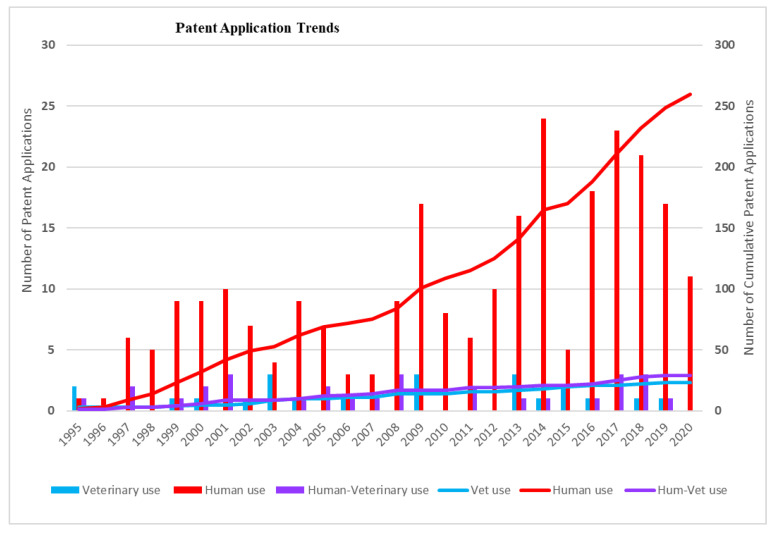
Patent applications for use in the human field (red line) are increasing, while patent applications for human–veterinary use (purple line) and veterinary use (blue line) are stabilizing. The bars indicate the number of patents per year, and the trendlines indicate the cumulative number of patent applications in this timeframe.

**Figure 2 pathogens-12-00046-f002:**
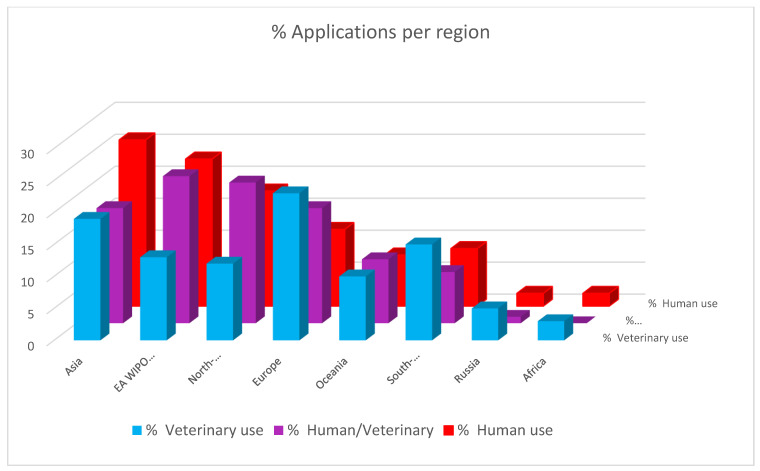
Asia shows the highest % of patent applications for human use. Followed by EA/WIPO/EPO and North America. Europe shows the highest % of patent applications is for veterinary use. The bars represent the % of patents per target group/per region.

**Figure 3 pathogens-12-00046-f003:**
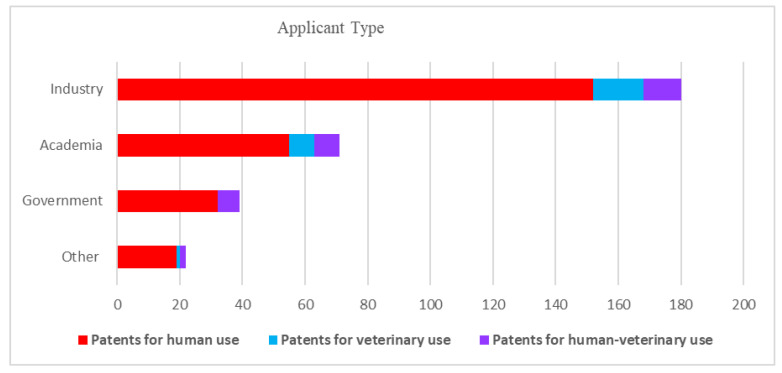
Industrial applicants dominate the field of patents for human use, followed by academic applicants.

**Figure 4 pathogens-12-00046-f004:**
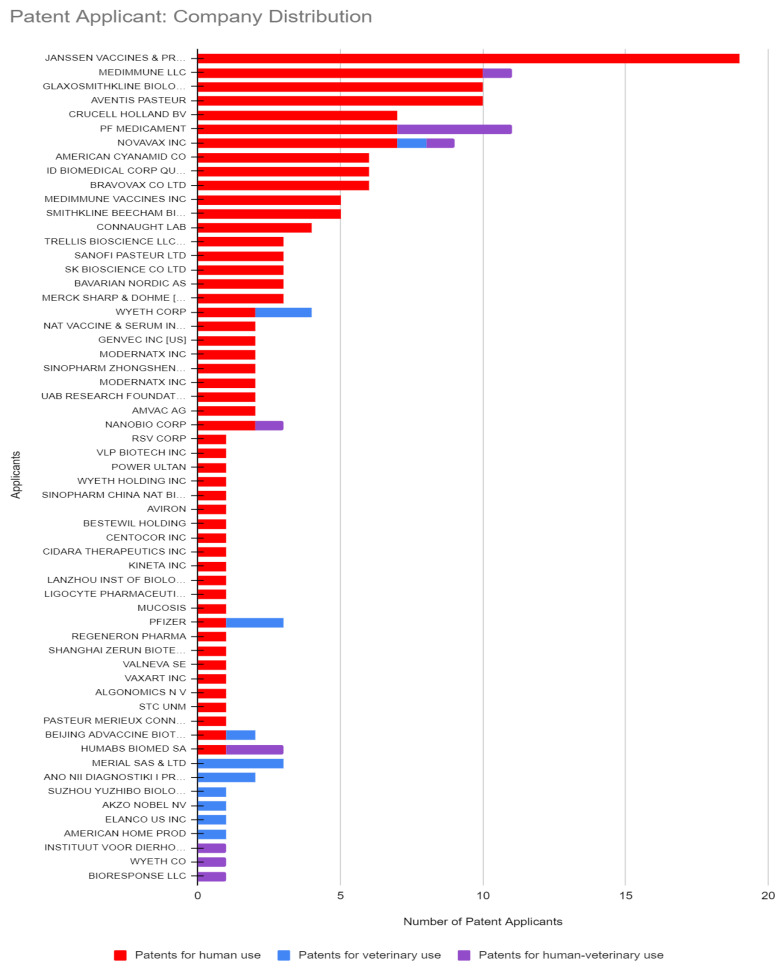
Janssen vaccines holds the most patents for human use. Novavax Inc is holder of patents in all three categories.

**Figure 5 pathogens-12-00046-f005:**
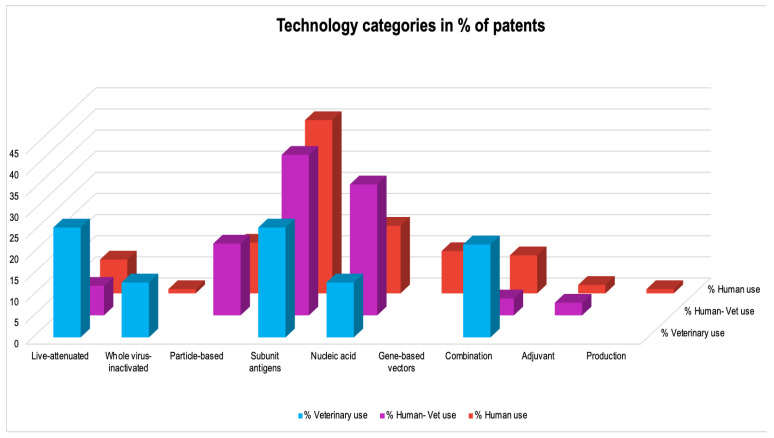
The majority of the technological inventions describe subunit vaccines, for all three target fields, of which most are using RSV F proteins, followed by inventions describing nucleic-acid technology. The bars represent the % of patents per target field/per technique.

**Table 1 pathogens-12-00046-t001:** CPC codes and keyword combinations used for patent search.

Field	CPC Codes and Keywords		Number of Patent Applications
Human	respiratory syncytial virus/RSV, vaccin* + immun* +	A61K39/155	1432
		C07K14/135	184
		C07K16/1027	129
		C12N2760/18511	39
		C12N2760/18521	5
		C12N2760/18522	17
		C12N2760/18534	170
		C12N2760/18543	242
		C12N2760/18561	22
		C12N2760/18564	22
		C12N2760/18571	3
		A61K2039/552	3
		C12N2760/18634	54
Veterinary	respiratory syncytial virus/RSV, vaccin* + immun* +	Y10S424/813	21
		A61K2039/552	3
		C12N2760/18634	54
For both veterinary and human use	respiratory syncytial virus/RSV, vaccin* + immun* +	A61K39, A61K31	32
Inclusion criteria	Patent documents focusing on the development of RSV vaccines targeting human, bovine, or human-bovine.		
Exclusion criteria	Not focused on RSV (e.g., parainfluenza virus, cancer, and coronavirus) Not describing vaccines, adjuvants or methods targeting RSV Not involving a human or bovine as target population		

## Data Availability

Not applicable.

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
