# Peer review of "Identifying Cross-Utilization of RSV Vaccine Inventions across the Human and Veterinary Field"

_pathogens, 2022, doi:10.3390/pathogens12010046_

Round 1

Reviewer 1 Report

Title: Identifying Cross-Utilization of RSV Vaccine Inventions Across the Human and Veterinary Field

The manuscript presented by Janse et al focused on cross-utilization of RSV vaccine innovations and collaboration across the human and veterinary fields. The knowledge sharing between human and veterinary fields is taking place benefiting the development of new vaccines in both fields. This information will help follow-up scientists/Researchers and/or policy makers on challenges they have encountered in human and veterinary fields. The article is well-written and organized, however, the following comments need to be addressed.

1.     For introduction info, use below reference instead of reference 6

Li et al. Global, regional, and national disease burden estimates of acute lower respiratory infections due to respiratory syncytial virus in children younger than 5 years in 2019: a systematic analysis. Lancet 2022; 399: 2047-64.

2.     What does it mean “ Not human”? make it easy so that non-scientist can understand.

3.     Can the author identify why the variation? Is it due to the effectiveness/efficacy of technologies? Or is the variability of RSV vaccine development due to the geographical distribution of applications or technologies or disease association enhancement? Please explain clearly in the discussion.

4.     Include Limitations/challenges of Cross-Utilization of RSV Vaccine in both filed in the text.

5.     Make a summary based on age category, target populations and technologies approaches in a tabular form for Cross-Utilization of RSV Vaccine in both human and veterinary filed.

Author Response

Dear reviewer 1,

Thank you very much for your time reading our paper and providing us with feedback.

Please below find our responses to the comments you provided us with.

*1.

Reference 6 is changed according to your suggestion, thank you for this adjustment.

*2.

Thank you very much for your question: what is the meaning of "not human" in Table 1. We adjusted the table describing 'not human' as 'For both veterinary and human use'. We adjusted the number of patents in line 133.

*Remark 3

Can the author identify why the variation? Is it due to the effectiveness/efficacy of technologies? Or is the variability of RSV vaccine development due to the geographical distribution of applications or technologies or disease association enhancement? Please explain clearly in the discussion.

As innovation across life sciences and Pharma Nutrition sectors critically depends on patenting as means to protect novel intellectual capital, patent documents can be used to signal these developments and identify stakeholders active in the field. Our paper identified a number of variations.  

  • First of all, a variation in yearly patent applications, (e.g. 2015 applications for human use are only 20% of the 2014 applications). Explanations for such variations could be found in epidemic numbers, however for this research we didn’t combine our findings to specific data for disease epidemiology and we cannot speculate on whether this is of any influence.
  • The total number of patents per geographical area also show variations, but on average this should reflect the expected markets. We have clarified this in lines 324-328.
  • As explained in the Discussion, variations in technology categories are linked to efficacy and safety concerns with different technology platforms (lines 357-362 and 369-381).

*Remark 4

Include limitations/ challenges cross-use RSV vaccine development

Creating a patent landscape is a challenging process and is never 100% accurate. The descriptions of the innovative inventions become public 18 months after the application of the patent. Meaning that our timeframe is missing the most recent inventions that are not yet public. We added a short remark on this in lines 302-305.

*Remark 5

Make a summary based on age category, target populations and technologies approaches in a tabular form for Cross-Utilization of RSV Vaccine in both human and veterinary filed.

We agree with the reviewer that such a table would be highly informative. Unfortunately, patent data is not amenable for such detailed information, as patents are often filed on inventions or technologies for which specific use (e.g. age category, target populations) are made specific in a later stage. The broad use or indicative applications and technology platforms are however mentioned in patents (human/ veterinary use), as this is a mandatory requirement for a patent document to be reviewed by the patent authorities. The combination of these latter two types of data (broad indicative use and technology) is shown in figure 5. We indicated that future research being in-depth follow-up studies on technologies and use should provide such an overview. Please see lines 417-424 and 439-444.

Reviewer 2 Report

Janse et al present a manuscript that assesses trends and breakthroughs in RSV vaccine development during 1995-2020, for the purpose of examining the extent of cross-utilization of inventions and the extent of collaborations across the human and bovine RSV fields. Human and bovine RSV are genetically closely related and also have similar disease patterns in aged and young humans and animals respectively. Furthermore, both have an unmet need for safe and efficacious vaccines. Therefore these viruses constitute a useful target to study cross-utilization, especially in light of global One-health efforts.

The authors used Espacenet to identify close to 1500 patent applications in this space. These applications were then subjected to various categorizations to identify trends. To do so, three target fields were distinghuished, human, veterinary, and human veterinary (the latter when both were mentioned as target groups in patent applications).

Although there is always some arbitrary element to categorizations, the work provides a nice overview of patents and ongoing and past collaborations, and a fair number of interesting and helpful factoids, relationships, and quantifications thereof, are presented. Some examples are the conclusions that Asia and North America have more patents in the human space whereas Europe has more in the veterinary space; that human patents increase whereas veterinary ones are stable and barely increasing; that there is more variation in vaccine technology within human patents than within veterinary patents; that cooperation for knowledge sharing is very limited between industry and academia and between the three target fields. Overall the paper concludes that despite the genetic and pathologic similarities, and the fact that both have unmet vaccine needs, there is little cross-utilization and the human and veterinary fields largely follow their separate processes in developing technology and vaccines.

In short, this manuscript is a good effort to attempt to document the extent of cross-utilization between human and animal viruses, in the light of worldwide One-health interests, and briefly discusses how One-health approaches may help develop strategies for potentially cost-saving sharing of knowledge and experience between the human and veterinary fields.

Author Response

Dear reviewer,

We thank you for your kind words and the concise summary of our paper.

Due to suggestions from reviewer 1 some minor adjustments are made. Table 1 is updated and we included extra info on the challenges of cross-use of patents in lines 302- 305 and a minor a spelling check has been carried out.

Kind regards,

Marga Janse